# Association of Thoracic Skeletal Muscle Index with Clinical Outcome and Response to Nutritional Interventions in Patients at Risk of Malnutrition—Secondary Analysis of a Randomized Trial

**DOI:** 10.3390/nu15040817

**Published:** 2023-02-05

**Authors:** Leonie Mueller, Nicole Mentil, Nathalie Staub, Stephanie Griot, Tobias Olpe, Felice Burn, Sebastian Schindera, Beat Mueller, Philipp Schuetz, Zeno Stanga, Annic Baumgartner

**Affiliations:** 1Medical Faculty of the University of Berne, 3010 Bern, Switzerland; 2Department Klinische Forschung (DKF), University of Basel, 4001 Basel, Switzerland; 3Department of Radiology Kantonsspital Aarau, 5000 Aarau, Switzerland; 4Medical University Department of Medicine, Kantonsspital Aarau, 5000 Aarau, Switzerland; 5Division of Diabetology, Endocrinology, Nutritional Medicine & Metabolism, Inselspital, Bern University Hospital, University of Bern, 3010 Bern, Switzerland

**Keywords:** computed tomography, sarcopenia, skeletal muscle, death, outcome, malnutrition, nutritional risk

## Abstract

Background: Measurement of skeletal muscle index (SMI) in computed tomography has been suggested to improve the objective assessment of muscle mass. While most studies have focused on lumbar vertebrae, we examine the association of SMI at the thoracic level with nutritional and clinical outcomes and response to nutritional intervention. Methods: We conducted a secondary analysis of EFFORT, a Swiss-wide, multicenter, randomized trial. We investigated the association of low SMI at the 12th thoracic vertebra (T12) with adverse outcome within 30 days after hospital admission (primary endpoint). Results: 663 of 2028 patients from the EFFORT trial had available CT scans for T12, and 519 among them also had available L3 scans. Mean SMI at T12 was 22.4 ± 5.8 cm^2^/m^2^ and 19.6 ± 5.5 cm^2^/m^2^ in male and female patients, respectively, and correlated well with nutritional parameters, including nutritional risk based on NRS 2002 (adjusted coefficient −0.63, 95%CI −1.25 to −0.01, *p* = 0.047), BMI (adjusted coefficient 0.74, 95%CI 0.66 to 0.82, *p* < 0.001) and handgrip strength (adjusted coefficient 0.15, 95%CI 0.11 to 0.2, *p* < 0.001). In multivariate regression analyses, low SMI was not a significant predictor for either clinical outcome or for treatment response. Results for SMI measured at L3 were similar, with only little prognostic value. Conclusions: Within medical patients at risk for malnutrition, SMI at thoracic vertebra provided low prognostic information regarding clinical outcomes and nutritional treatment response.

## 1. Introduction

Sarcopenia is defined as a reduction in either muscle mass or quality in the context of impaired muscle function and has been shown to predict adverse outcome in different patient populations, particularly in patients with malnutrition [1,2,3,4]. In the context of malnutrition, the recently developed GLIM (Global Leadership Initiative on Malnutrition) criteria suggest integration of sarcopenia as a core component of diagnostic workup [5]. Yet, there is uncertainty regarding the best definition of sarcopenia [6,7].

Indeed, there is a need to better validate the different available tools to assess muscle mass and muscle function, including different imaging modalities as well as functional tests. Yet, computer tomography (CT)-based diagnosis of sarcopenia has emerged as a reliable and objective method to measure muscle mass. Herein, CT-based measures allow the assessment of both skeletal muscle and adipose tissue. For quantification of muscle mass, previous researchers have suggested the use of the skeletal muscle index (SMI), which estimates the area of total skeletal muscle (cm^2^) in relation to height squared (m^2^) [5,6,8]. Skeletal muscle mass at L3 has correlated well with whole body muscle mass and with clinical outcomes in several studies [1,9,10,11,12] and in a previous analysis of our patient cohort. Still, the main disadvantage is radiation exposure, which limits the usefulness of CT as a primary screening tool. Yet, due to frequent use of CT scans in clinical routine, particularly in hospitalized patients, the use of these scans for assessing muscle mass is an intriguing possibility to gain clinically relevant information. However, the third lumbar vertebra is only available on abdominal and abdomino-pelvic CT scans, and a large portion of patients may only receive thoracic CT scans in routine care, i.e., for exclusion of pulmonary embolism and for assessing lung infection. So far, only a few studies have evaluated the correlation of skeletal muscle mass in thoracic and abdominal CT scans [8,13,14,15], and the reliability and predictive value regarding clinical outcomes remains understudied.

Herein, our aim was to examine the association of SMI at level T12 with clinical outcomes as well as nutritional outcomes and with the response to nutritional intervention in patients included in the Effect of early nutritional support on Frailty, Functional Outcomes, and Recovery of malnourished medical inpatients Trial (EFFORT) [16].

## 2. Materials and Methods

### 2.1. Study Design and Setting

We conducted a post hoc, secondary analysis of the randomized-controlled, open-label, 8-centre EFFORT trial [16]. Several hospitals in Switzerland recruited patients for this trial, including the University and Kantonalspital in Bern, Aarau, Lucerne, Solothurn, St. Gallen, Münsterlingen and Baselland and Lachen. In these hospitals, malnutrition screening was established with the nutritional risk screening 2002 (NRS 2002) score [17]. Patients with a score of ≥3 points are considered to be at nutritional risk.

The trial was approved by the ethic committee of Northwestern Switzerland (EKNZ; 2014_001) and registered in ClinicalTrials.gov (https://clinicaltrials.gov/ct2/show/NCT02517476 (accessed on 7 August 2015)).

### 2.2. Patient Population

The inclusion criteria for the EFFORT study included a NRS score ≥ 3 points, age over 18 years, an expected length of hospital stay of ≥4 days and the informed consent of the patients. We had several exclusion criteria, including patients from the ICU or surgical patients, patients with anorexia nervosa, terminal illness, specific illnesses such as pancreatitis, and liver failure, as well as stem cell transplantation or cystic fibrosis. We also excluded patients with a previous history of gastric bypass surgery and patients that were not able to ingest oral nutrition or had ongoing nutritional support and/or allergies or contraindications for nutritional support. After giving informed consent, patients were randomized in a 1:1 ratio to the intervention group to receive individualized nutritional support or the control group to receive standard hospital food.

For the present secondary analysis, 663 patients of the original EFFORT trial who received an abdominal, abdomino-pelvic or thoracic CT scan containing level T12 within 3 months of trial inclusion were eligible. Among these 663 patients, we also had 519 patients with available L3 scans that were used to compare results.

### 2.3. Nutritional Procedures during the Trial

Nutritional procedures during the trial are summarized in the study protocol [16]. In brief, randomization was done through an interacting web system, with variable block sizes and stratification according to the site and the severity of malnutrition. The intervention group received individualized nutritional support according to an implementation protocol [18], while the control group received standardized hospital food without nutritional support. Energy requirements were calculated using the Harris–Benedict equation [19]. Protein requirements were set at 1.2–1.5 g/kg body weight [20]. We defined lower targets for renal failure patients, with 0.8 g protein per kg body weight. For each patient, we developed, in collaboration with the dietician team, an individual nutrition plan to reach these goals [21,22]. Control group patients had usual care hospital food, without additional counseling.

### 2.4. Image Review and Evaluation

In a first step, centrally trained researchers assessed the quality of the original Digital Imaging and Communications in Medicine (DICOM) images at level T12 and L3. In a second step, the research assistants evaluated the CT scans and selected a single slice at level T12. For this purpose, we used SliceOmatic Software version 5.0 (TomoVision, Montreal, Quebec, QC, Canada). We excluded all images with incomplete depiction of T12 and in case that the muscle tissue was out of range and/or if contrast did not allow discrimination. There were no anatomical variations that led to exclusion.

Muscle and visceral tissue were distinguished from subcutaneous adipose tissue using tissue-specific Hounsfield Unit (HU) ranges and anatomical knowledge. We followed the Alberta protocol [23] and set Hounsfield ranges to −29 to 150 HU for skeletal muscle, −190 to −30 HU for subcutaneous and intramuscular adipose tissue, and −150 to −50 HU for visceral adipose tissue. Muscles included in the cross-sectional measurements at T12 with different muscle groups, including the erector spinae, latissimus dorsi, external and internal oblique, rectus abdominis and external and internal intercostal muscles. Every slice was evaluated twice to improve the interrater reliability with the aim to achieve >99%. Researchers improved the slicing and measuring criteria after the first round.

### 2.5. Quantification of Muscle Mass

For the quantification of muscle mass, the skeletal muscle index (SMI) was used, which is calculated from the total muscle area at level T12, divided by the patient height (m^2^). Because there are no internationally accepted cut-off values for the diagnosis of sarcopenia based on thoracic level CT scans, we defined low SMI for patients with an SMI within the lowest sex-specific SMI quartile. The cut-off for females was 30.6 cm^2^/m^2^ and for males 42.6 cm^2^/m^2^. For comparison with lumbar spine CTs, we used a similar approach for L3 measurements, comparing patients in the lowest quartile to patients in the three higher quartiles.

### 2.6. Clinical Outcomes

We defined the outcomes for this analysis in accordance with the EFFORT trial. Specifically, the primary composite endpoint was defined as adverse outcome after hospital admission within 30 days [16]. This included all-cause mortality, intensive care unit admission, major complications, rehospitalization and a functional decline from baseline to day 30 within this 30 day time frame. We also had several additional short- and long-term outcomes, including all-cause mortality within 30 days and 180 days, readmission to hospital care within 30 days, length of hospital stay and functional decline within 30 days. Rehospitalization was considered as a non-elective admission to hospital care within 30 days after discharge. Functional decline was measured by the Barthel’s Index. The cut-off for a decrease was defined as a reduction of 10% within 30 days. Study nurses blinded to the randomization assessed endpoints by structured interviews via phone calls. When necessary, the survival status was confirmed by contacting the patient’s general practitioner or family members.

### 2.7. Statistical Analyses

For the assessment of the predictive factors of SMI, we used a linear regression model. For the multivariate regression, we adjusted for the confounders C-reactive protein (CRP) and serum albumin, as well as for handgrip strength. Pearson correlation was used to compare thoracic and lumbar SMI. To examine associations of SMI at T12 with clinical outcome, we used a logistic regression model for binary outcomes and linear regression models for continuous variables. The multivariate regression calculations were adjusted for age, BMI, nutritional support intervention, contributing center, and presence of major comorbidities, i.e., stroke, COPD, hypertension, diabetes and chronic heart. Furthermore, an analysis with subgroups to investigate differences in specific patient groups (age ≥ 80 years, NRS score ≥ 4 points, male gender and the presence of a tumor or frailty) was conducted. We used logistic regression to investigate associations with stratification by SMI. We used Stata 15.1 Software (Stata Corp, College Station, TX, USA). Statistical significance for two-sided tests was set for *p*-values < 0.05.

## 3. Results

### 3.1. Patient Population

Of the 2028 EFFORT trial patients, 663 (32.7%) had a thoracic, abdominal or abdomino-pelvic CT scan with T12 available and were included in this study (Table 1). Among those, 519 patients also had available L3 scans. A total of 294 (44.3%) patients were female, and the mean age was 70.5 (±13.3) years. The mean SMI at T12 was 22.4 cm^2^/m^2^ (±5.8) for males and 19.6 cm^2^/m^2^ (±5.5) for females. There was a strong correlation between SMI at level L3 and T12 (r = 0.74, *p* < 0.001). A total of 167 (25%) patients had a low sex-specific SMI based on our definition. Patients with low SMI had a significantly lower BMI and body weight, a higher nutritional risk based on NRS and lower handgrip strength (22.6 kg vs. 26.1 kg). There were no differences in regard to main diagnoses or comorbidities (Table 1).

### 3.2. Association of Low SMI and Clinical Markers

In a first step, we investigated the association of SMI with different nutritional markers (Table 2). We found a positive association of SMI with weight, with an increase in the SMI of 0.22 per one kilogram higher weight (95%CI 0.2–0.24; *p* < 0.001). This association was also robust when adjusting for different confounding factors, including for albumin, C-reactive protein (CRP) and handgrip strength (adjusted coefficient 0.23, 95%CI 0.20 to 0.25, *p* < 0.001). A similar association was also found for BMI, NRS and handgrip strength, with significant results in the unadjusted and adjusted analyses. The area under the curve for all parameters, however, suggested only low to moderate discrimination.

### 3.3. Association of Low SMI and Clinical Outcomes

In a second step, we investigated the association of low SMI with different clinical outcomes (Table 3). The risk for adverse outcome was similar in patients with high SMI (135/496, 27.2%) compared to patients with low SMI (52/167, 31.1%), resulting in a non-significant adjusted odds ratio (OR) of 1.37 (95%CI 0.89 to 2.11, *p* = 0.157). Similarly, there were no significant associations for most other secondary endpoints, including all-cause mortality at 30 and 180 days. The same analysis was also repeated with SMI as a continuous variable, with very similar results (Table 3, right column).

In addition, we also investigated the value of low SMI measured at L3 (current standard) with the same clinical outcomes (Appendix A). Similar to TH12, the prognostic value of L3 was low in regard to ORs and AUCs.

We also performed a subgroup analysis stratifying patients based on their risk for low SMI (i.e., patients ≥80 years of age and patients with NRS ≥ 4), which showed similar, mostly non-significant results. The results of the subgroup analyses are presented in the appendix (Appendix A).

### 3.4. Association of Low SMI and Response to Nutritional Support

In a final step, we investigated whether SMI may help to predict response to nutritional treatment. We compared differences in clinical outcomes among intervention group and control group patients according to low or high SMI measurements (Table 4). In regards to adverse clinical outcome within 30 days (primary endpoint), the OR for the nutritional support intervention was similar in the group of patients with high SMI compared to patients with low SMI, with no significant results in the interaction analysis. Results were similar for all endpoints, without evidence for effect modification.

The same analysis was also repeated stratifying patients as high or low SMI according to L3 measurements (Appendix A). Again, results did now show evidence that low SMI at L3 would predict treatment response.

## 4. Results

The main results of this secondary analysis of a randomized trial investigating the association of low thoracic skeletal muscle mass with different clinical and nutritional outcomes and with the response to nutritional treatment in patients at nutritional risk are as follows. First, we found significant and independent associations of the SMI with different nutritional parameters, including BMI, nutritional risk as assessed by the NRS 2002 and handgrip strength, suggesting that thoracic SMI is an additional nutritional parameter that may help to better characterize patients and identify patients at nutritional risk based on a routine examination. Second, the prognostic implications of low thoracic skeletal muscle mass were only moderate, with non-significant results in an adjusted regression analysis and with low area under the curve values. Third, there was little evidence that low thoracic skeletal muscle mass would help to identify patients that show a more pronounced response to nutritional support.

Our study is important in regard to the current discussion about the use of GLIM criteria to diagnose malnutrition [5,24]. The Global Leadership Initiative on Malnutrition (GLIM) recently proposed specific criteria for the diagnosis of malnutrition [5,24]. GLIM proposes a relatively straightforward two-step approach. First, patients should undergo screening to identify patients at risk of malnutrition. This is followed by a more in-depth assessment, with more specific criteria to diagnose malnutrition [5,24]. Historically, several criteria have been proposed but have lacked sensitivity and specificity [25]. The proposed GLIM criteria should provide more specific criteria to diagnose malnutrition. These include three phenotypic criteria and two etiological criteria [7]. To diagnose malnutrition, one phenotypic criterion and one etiologic criterion must be present. However, while the prognostic validity of GLIM is established, it remains unclear whether these criteria can be helpful to guide treatment [26]. Herein, individual markers of muscle health may help to select patients regarding treatment [27]. Still, this analysis found little value in thoracic CT scans for this purpose.

Today, there is limited research on the usefulness of the SMI in thoracic CT scans to predict clinical outcomes. Nemec et al. showed an association of low SMI measured at T12 with longer hospital stays in a population of patients undergoing transcatheter aortic valve replacement (TAVR) [8]. Olson et al. found an increased risk of mortality in patients with thoracic CT-based sarcopenia undergoing thoracic endovascular aortic repair (TEVAR) compared with non-sarcopenic subjects [14]. However, in the study by Olson, a different method was used for assessing skeletal muscle mass compared to our study, normalizing the cross-sectional area by total body area using the Mosteller formula, rather than normalizing by height. Miller et al. examined skeletal muscle mass of the erector spinae muscles and the pectoralis muscles separately in patients receiving lobectomy [28]. The erector spinae SMI, measured at T12, was associated with lower survival after 30 days and prolonged length of stay, which was not the case for SMI of the pectoralis muscles. Tanimura et al. showed similar results in a population of patients with COPD, with higher mortality risk in patients with low erector spinae muscle at T12 [29]. Moon et al. found an association of low SMI at T4 and T12 level with higher mortality; however, after adjustment for confounders, the results for the measurements at T12 were no longer statistically significant. These results and associations with different clinical outcomes may in part be explained by the wide variety in study methodology and differences in the assessment of muscle mass. Herein, we believe our results of a relatively large and well-characterized cohort of medical inpatients at nutritional risk is important and suggests only little additional value of thoracic CT scans to predict outcomes.

Currently, there is no well-defined cut-off value for SMI at level T12 for the diagnosis of sarcopenia. While some studies have suggested cut-off values based on their own patient populations, using these cut-offs in our cohort did not match well with the population, and >95% of patients would have been classified as having low SMI [8,13,15]. This difference may be explained by higher age and higher frequencies of frailty and comorbidities among the EFFORT population, while Nemec et al. and Olson et al. included patients with a predominantly cardiovascular risk profile, and Derstine et al. examined a healthy population. Importantly, this shows the need to validate cut-offs within the population of patients where an examination is being done.

Interesting, in the EFFORT study, there were more routine CT scans with T12 done compared to L3 CT scans (i.e., 663 vs. 519, respectively), suggesting that in clinical routine, thoracic scans may be more widely available in this population of medical inpatients. Still, our results do not support the use of the SMI in single cross-sectional images at level T12 for the definition of clinically relevant sarcopenia due to the lack of prognostic information derived from these measurements. Of note, L3 measurements also provided very little prognostic information in this analysis.

This study has some strengths and limitations. EFFORT was based on a prospective, randomized, multicenter study, and therefore the population included for analyses was large and well characterized. To our knowledge, this was the first study to investigate an association between low SMI on thoracic CT scans with response to nutritional support in patients at nutritional risk. We have previously reported results of L3 measurements in the overall cohort and results may differ from this report that was limited to patients with available Th12 scans [30]. The Main limitation includes the limited power of the analysis due to no consecutive performance of CT scans and a risk of selection bias.

## 5. Conclusions

Within this cohort of medical patients at risk for malnutrition, skeletal muscle index measured at the thoracic vertebra provided low prognostic information regarding clinical outcomes and nutritional treatment response. There is a need for similar research and other patient populations to understand the added value of skeletal muscle index measured at the thoracic vertebra for patient assessment.

## Figures and Tables

**Table 1 nutrients-15-00817-t001:** Baseline characteristics overall and stratified by SMI.

Characteristic	Overall	High SMI	Low SMI	*p* Value
	(*n* = 663)	(*n* = 496)	(*n* = 167)	
Socio-demographics				
Age, mean (SD)	70.5 (13.3)	70.4 (13.3)	70.9 (13.6)	0.70
Biological sex—Male	369 (55.7%)	276 (55.6%)	93 (55.7%)	0.99
Nutritional history				
BMI (kg/m^2^), mean (SD)	25.2 (5.0)	26.3 (4.9)	21.7 (3.4)	<0.001
Body weight (kg), mean (SD)	72.3 (16.3)	75.8 (16.3)	62.3 (11.2)	<0.001
NRS, mean (SD)	4.08 (0.89)	4.02 (0.88)	4.25 (0.90)	0.003
NRS 2002 score = 3	203 (30.6%)	162 (32.7%)	41 (24.6%)	0.012
NRS 2002 score = 4	242 (36.5%)	188 (37.9%)	54 (32.3%)	
NRS 2002 score = 5	182 (27.5%)	121 (24.4%)	61 (36.5%)	
NRS 2002 score = 6	36 (5.4%)	25 (5.0%)	11 (6.6%)	
Weight loss				0.39
≤5% in 3 months	321 (48.4%)	241 (48.6%)	80 (47.9%)	
>5% in 3 months	88 (13.3%)	63 (12.7%)	25 (15.0%)	
>5% in 2 months	95 (14.3%)	77 (15.5%)	18 (10.8%)	
>5% in 1 month	159 (24.0%)	115 (23.2%)	44 (26.3%)	
Loss of appetite within the last 30 days				0.71
No	78 (11.8%)	57 (11.5%)	21 (12.6%)	
Yes	585 (88.2%)	439 (88.5%)	146 (87.4%)	
Food intake of normal requirement preceding week—no (%)				0.10
>75%	64 (9.7%)	48 (9.7%)	16 (9.6%)	
50–75%	212 (32.0%)	163 (32.9%)	49 (29.3%)	
25–50%	277 (41.8%)	213 (42.9%)	64 (38.3%)	
<25%	110 (16.6%)	72 (14.5%)	38 (22.8%)	
Severity of illness—no (%)				0.71
very mild	12 (1.8%)	10 (2.0%)	2 (1.2%)	
mild	386 (58.2%)	286 (57.7%)	100 (59.9%)	
moderate	257 (38.8%)	195 (39.3%)	62 (37.1%)	
severe	8 (1.2%)	5 (1.0%)	3 (1.8%)	
CRP mean (SD)	8.24 (9.12)	8.14 (8.93)	8.54 (9.68)	0.63
Albumin mean (SD)	29.61 (6.63)	29.78 (6.37)	29.16 (7.32)	0.36
Muscle mass				
T12 Skeletal Muscle Index in males in cm^2^/m^2^; mean (SD)	22.44 (5.79)	24.70 (4.85)	15.73 (1.72)	<0.001
T12 Skeletal Muscle Index in females cm^2^/m^2^; mean (SD)	19.61 (5.51)	21.65 (4.82)	13.53 (1.45)	<0.001
Handgrip strength mean (SD)	25.2 (11.8)	26.1 (12.4)	22.6 (9.6)	0.002
Main admission diagnosis, n (%)				
Infection	192 (29.0%)	146 (29.4%)	46 (27.5%)	0.64
Oncologic disease	215 (32.4%)	156 (31.5%)	59 (35.3%)	0.35
Cardiovascular disease	38 (5.7%)	30 (6.0%)	8 (4.8%)	0.55
Frailty	59 (8.9%)	42 (8.5%)	17 (10.2%)	0.50
Lung disease	40 (6.0%)	27 (5.4%)	13 (7.8%)	0.27
Gastrointestinal disease	50 (7.5%)	39 (7.9%)	11 (6.6%)	0.59
Neurological/psychiatric disease	14 (2.1%)	11 (2.2%)	3 (1.8%)	0.74
Renal disease	14 (2.1%)	11 (2.2%)	3 (1.8%)	0.74
Metabolic disease	13 (2.0%)	12 (2.4%)	1 (0.6%)	0.14
Other	14 (2.1%)	12 (2.4%)	2 (1.2%)	0.34
Comorbidities, n (%)				
Hypertension	353 (53.2%)	272 (54.8%)	81 (48.5%)	0.16
Tumor	326 (49.2%)	239 (48.2%)	87 (52.1%)	0.38
Renal failure	175 (26.4%)	135 (27.2%)	40 (24.0%)	0.41
Coronary heart disease	163 (24.6%)	125 (25.2%)	38 (22.8%)	0.53
Diabetes mellitus	120 (18.1%)	98 (19.8%)	22 (13.2%)	0.056
Chronic heart failure	82 (12.4%)	62 (12.5%)	20 (12.0%)	0.86
COPD	96 (14.5%)	74 (14.9%)	22 (13.2%)	0.58
Peripheral artery disease	39 (5.9%)	27 (5.4%)	12 (7.2%)	0.41
Stroke	42 (6.3%)	34 (6.9%)	8 (4.8%)	0.34
Dementia	12 (1.8%)	9 (1.8%)	3 (1.8%)	0.99

Abbreviations: SMI= skeletal muscle index; SD = standard deviation; BMI = body mass index; NRS 2002 = nutritional risk screening 2002; COPD = chronic obstructive pulmonary disease. We defined low SMI as the lowest quartile of SMI and high SMI as the other three quartiles of SMI of this study population.

**Table 2 nutrients-15-00817-t002:** Association of SMI with clinical markers.

Parameter	Univariate Regression	ROC Area	Female	Male	*p Interaction*	Multivariate Regression Adjusted for Albumin, C-Reactive Protein (CRP), Handgrip Strength
	Coefficient (95%CI) *p* value		Coefficient (95%CI) *p* value	Coefficient (95%CI) *p* value		Coefficient (95%CI) *p* value
Nutritional marker						
NRS, per point increase	−0.82 (−1.32 to −0.33), *p* = 0.001	0.57	−0.32 (−1.03 to 0.39), *p* = 0.373	−1.25 (−1.9 to −0.59), *p* < 0.001	0.381	−0.63 (−1.25 to −0.01), *p* = 0.047
Weight, per kg	0.22 (0.2 to 0.24), *p* < 0.001	0.25	0.2 (0.16 to 0.24), *p* < 0.001	0.23 (0.2 to 0.26), *p* = 0.00	<0.001	0.23 (0.2 to 0.25), *p* < 0.001
Weight loss (refers to 4 categories: (≤5% in 3 month, >5% in 3 month, <5% in 2 month, <5% in 1 month)	−0.32 (−0.67 to 0.04), *p* = 0.081	0.50	−0.2 (−0.7 to 0.3), *p* = 0.434	−0.59 (−1.06 to −0.11), *p* = 0.015	0.445	−0.39 (−0.82 to 0.04), *p* = 0.076
BMI, per 1 unit increase	0.71 (0.64 to 0.78), *p* < 0.001	0.20	0.58 (0.49 to 0.67), *p* < 0.001	0.84 (0.75 to 0.94), *p* < 0.001	0.000	0.74 (0.66 to 0.82), *p* < 0.001
Clinical marker						
Handgrip strength	0.13 (0.1 to 0.17), *p* < 0.001	0.43	0.04 (−0.06 to 0.14), *p* = 0.43	0.11 (0.06 to 0.16), *p* < 0.001	0.433	0.15 (0.11 to 0.2), *p* < 0.001
Loss of appetite	0.98 (−0.4 to 2.36), *p* = 0.165	0.49	1.1 (−0.99 to 3.19), *p* = 0.302	1.21 (−0.55 to 2.97), *p* = 0.176	0.315	1.62 (−0.07 to 3.31), *p* = 0.061
Food intake (>75%, 50–75%, 25–50%, <25% of normal requirement preceding week)	−0.17 (−0.68 to 0.35), *p* = 0.523	0.54	−0.33 (−1.1 to 0.44), *p* = 0.396	0.03 (−0.63 to 0.69), *p* = 0.933	0.409	−0.01 (−0.65 to 0.63), *p* = 0.978
Disease severity (very mild, mild, moderate, severe)	0.23 (−0.58 to 1.05), *p* = 0.577	0.50	0.08 (−1.03 to 1.2), *p* = 0.882	0.18 (−0.94 to 1.3), *p* = 0.75	0.885	0.16 (−0.9 to 1.22), *p* = 0.763
Blood marker						
Albumin, per 1 g/dL	0.05 (−0.03 to 0.13), *p* = 0.232	0.48	−0.03 (−0.13 to 0.08), *p* = 0.603	0.12 (0.01 to 0.23), *p* = 0.026	0.617	0.03 (−0.06 to 0.12), *p* = 0.541
CRP (mg/L, per 10 unit increase)	0 (0 to 0.01), *p* = 0.836	0.50	0.05 (−0.02 to 0.12), *p* = 0.13	−0.05 (−0.11 to 0.02), *p* = 0.173	0.138	0.00 (−0.01 to 0.01), *p* = 0.975

Abbreviations: NRS denotes nutritional risk screening 2002; BMI denotes body mass index; ROC denotes receiver operator characteristic curve.

**Table 3 nutrients-15-00817-t003:** Association of low SMI at level Th12 with clinical outcomes.

	High SMI	Low SMI		AUC	SMI at Level T12, Continuous	AUC
	*n* (%) of Patients with high SMI	*n* (%) of Patients with low SMI	OR or Coefficient (95%CI),		OR or Coefficient (95%CI)	
			*p* value, adjusted for age, BMI, nutritional support intervention, contributing center, presence of stroke, COPD, hypertension, diabetes, chronic heart failure		*p* value, adjusted for age, BMI, nutritional support intervention, contributing center, presence of stroke, COPD, hypertension, diabetes, chronic heart failure	
	*n* = 496	*n* = 167			*n* = 663	
Primary endpoint						
Adverse clinical outcome within 30 days	135 (27.2%)	52 (31.1%)	1.37 (0.89, 2.11), *p* = 0.157	0.52	−0.1 (−0.18, −0.02), *p* = 0.12	0.46
Short-term endpoints						
30-day all-cause mortality	45 (9.1%)	19 (11.4%)	1.65 (0.86, 3.18), *p* = 0.132	0.52	−0.004 (−0.009, 0.005), *p* = 0.08	0.48
Rehospitalization within 30 days	53 (10.7%)	19 (11.4%)	1.06 (0.57, 1.96), *p* = 0.864	0.51	−0.002 (−0.008, 0.002), *p* = 0.32	0.48
Mean length of stay, days (SD)	10.0 (6.9)	10.1 (7.5)	0.42 (−0.94, 1.78), *p* = 0.548	-	−0.008 (−0.13, 0.11), *p* = 0.89	-
Decline Barthel index score (points) after 30 days	68 (13.7%)	29 (17.4%)	1.75 (1.01, 3.05), *p* = 0.048	0.53	0.05 (−0.008, 0.1), *p* = 0.09	0.48
Long-term endpoints						
180-day all-cause mortality	147 (29.6%)	47 (28.1%)	1.08 (0.69, 1.7) *p* = 0.732	0.49	−0.002 (−0.01, 0.005) *p* = 0.53	0.52

Abbreviations: SD denotes standard deviation; OR denotes odds ratio; CI denotes confidence interval; AUC denotes area under the curve.

**Table 4 nutrients-15-00817-t004:** Effects of nutritional support on clinical outcomes of patients in the lowest and in the other three quartiles of SMI at level T12.

	High SMI			Low SMI			
	Control Group (*n* = 243)	Intervention Group (*n* = 253)	OR or Coefficient (95%CI), *p* Value Adjusted	Control Group (*n* = 79)	Intervention Group (*n* = 88)	OR or Coefficient (95%CI), *p* Value Adjusted	*p* for Interaction
Primary endpoint							
Adverse clinical outcome within 30 days	71 (29.2%)	64 (25.3%)	0.48 (0.23, 0.99), *p* = 0.048	29 (37%)	23 (26%)	0.84 (0.56, 1.26), *p* = 0.4	0.248
Short-term endpoints							
30-day all-cause mortality	25 (10.3%)	20 (7.9%)	0.4 (0.12, 1.29), *p* = 0.124	12 (15%)	7 (8%)	0.77 (0.41, 1.45), *p* = 0.424	0.362
Rehospitalization within 30 days	26 (10.7%)	27 (10.7%)	0.84 (0.29, 2.38), *p* = 0.737	9 (11%)	10 (11%)	1.01 (0.57, 1.8), *p* = 0.976	0.889
Length of hospital stay	10.0 (6.4)	10.1 (7.4)	−2.65 (−5.03, −0.28), *p* = 0.029	11.4 (7.6)	9.0 (7.1)	−2.65 (−5.03, −0.28), *p* = 0.029	
Decline Barthel index score	42 (17.3%)	26 (10.3%)	0.28 (0.1, 0.76), *p* = 0.013	18 (21%)	9 (11%)	0.58 (0.34, 0.99), *p* = 0.045	0.203
Long-term endpoints							
180-day all-cause mortality	75 (30.9%)	72 (28.5%)	0.63 (0.3, 1.32), *p* = 0.221	25 (32%)	22 (25%)	0.91 (0.6, 1.37), *p* = 0.648	0.303

Abbreviations: OR denotes odds ratio; CI denotes confidence interval. *p* values were adjusted for important confounders, including age, BMI, nutritional support intervention, contributing center, presence of stroke, COPD, hypertension, diabetes and chronic heart failure.

## Data Availability

Data described in the manuscript, code book, and analytic code will be made available upon request pending finalization of other secondary projects related to the trial.

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
