# Peer review of "Association of Thoracic Skeletal Muscle Index with Clinical Outcome and Response to Nutritional Interventions in Patients at Risk of Malnutrition—Secondary Analysis of a Randomized Trial"

_nutrients, 2023, doi:10.3390/nu15040817_

Round 1
Reviewer 1 Report
The proposed manuscript is based on secondary analysis of previously available data to study the relevance of measuring the skeletal muscle index to define sarcopenia at the T12 level, and not L3, as this data is more easily accessible in current practice. The optimization of health resources, as well as the definition of tools usable in current practice to define sarcopenia seem essential. The manuscript presents a negative result, which is unfortunate with regard to the relevance of the question asked, but which is essential for the progress of knowledge. This should be encouraged.
As the study builds on a previous protocol, the underlying methodological elements are clear. The manuscript is fluid and pleasant to read.
My only concern is this: if the data are available, can a comparison of the L3 and T12 data on the same cohort be conducted? Indeed, such an analysis would be an important addition to determine the added value, or not, of L3. If such an analysis is feasible, can the authors conduct it? If not, can they justify ?
Author Response
The proposed manuscript is based on secondary analysis of previously available data to study the relevance of measuring the skeletal muscle index to define sarcopenia at the T12 level, and not L3, as this data is more easily accessible in current practice. The optimization of health resources, as well as the definition of tools usable in current practice to define sarcopenia seem essential. The manuscript presents a negative result, which is unfortunate with regard to the relevance of the question asked, but which is essential for the progress of knowledge. This should be encouraged.
As the study builds on a previous protocol, the underlying methodological elements are clear. The manuscript is fluid and pleasant to read.
My only concern is this: if the data are available, can a comparison of the L3 and T12 data on the same cohort be conducted? Indeed, such an analysis would be an important addition to determine the added value, or not, of L3. If such an analysis is feasible, can the authors conduct it? If not, can they justify ?
Reply: thank you for your positive feedback. We have now added the data regarding L3 to the manuscript as suggested and because results were similar with low prognostic implications, we changed the conclusions in this regard.
Reviewer 2 Report
This paper investigated the association between thoracic SMI with prognosis and response to nutritional support, using the population of a multi-center cohort from EFFORT trial. Although this study included relatively large number of participants, the result itself is neither noteworthy nor interesting. It is unclear whether thoracic SMI is useful compared to that measured at L3.
Major concerns and questions for authors:
1. The most significant concern is that, as described above, the advantage and disadvantage of thoracic SMI compared to lumbar SMI are unclear from the paper, although the aim of this study included determining the usefulness of a more accessible thoracic SMI as an alternative to lumbar SMI. Not only the correlation analysis between SMI at level T12 and L3, the authors should also compared the predictive value of both directly in the analyses. Although the authors describe that “Thus, measurements at level L3 may be the preferred option,” this is not supported by the results.
2. In connection with the above, if clinical usefulness or meaningfulness of SMI at T12 and L3 is different, the authors should explain the reason in more detail.
3. Although the participants were divided into 2 groups (high SMI and low SMI) with a cutoff of lowest quartiles, are there linear correlations or non-linear correlations (e.g. cubic spline curve) between thoracic SMI and the risk of outcomes? It is interesting and meaningful to perform the analyses with SMI as a continuous variable, not only as a binary variable.
4. Regarding the analyses in Table 4, the background characteristics of control and intervention groups in high or low SMI groups are not considered randomized because this is the post-hoc analysis of RCT. Therefore, “adjusted” values are necessary, as same as those in Table 3.
5. In Table 4, what the “interaction” indicates? If it means interaction between intervention and SMI groups, it is surprising that there is a significant interaction despite of the similar OR in high and low SMI groups (0.55 vs. 0.40).
6. Regarding Supplementary Table 1, analyses should also be performed in the patients with NRS < 4 and age < 80 years. Additionally, interaction between SMI and subgroups should be calculated.
Minor concerns and questions for authors:
1. In the Abstract – please insert “(SMI)” after “skeletal mass index” in the first line.
2. In the Discussion section, the authors described that “in the EFFORT study, there were more routine CT scans with T12 done compared to L3 CT scan.” This statement is closely related to the aim of this study. Please clarify the number of patients performed T12 and L3 scans, respectively.
3. Additionally, the authors should describe whether the issue above can be generalized to other populations with references.
Author Response
This paper investigated the association between thoracic SMI with prognosis and response to nutritional support, using the population of a multi-center cohort from EFFORT trial. Although this study included relatively large number of participants, the result itself is neither noteworthy nor interesting. It is unclear whether thoracic SMI is useful compared to that measured at L3.
Reply: thank you for your positive feedback. We have now added the data regarding L3 to the manuscript as suggested but both reviewers.
Major concerns and questions for authors:
- The most significant concern is that, as described above, the advantage and disadvantage of thoracic SMI compared to lumbar SMI are unclear from the paper, although the aim of this study included determining the usefulness of a more accessible thoracic SMI as an alternative to lumbar SMI. Not only the correlation analysis between SMI at level T12 and L3, the authors should also compared the predictive value of both directly in the analyses. Although the authors describe that “Thus, measurements at level L3 may be the preferred option,” this is not supported by the results.
Reply: thank you – we agree and have now added the data regarding L3 to the manuscript as suggested.
- In connection with the above, if clinical usefulness or meaningfulness of SMI at T12 and L3 is different, the authors should explain the reason in more detail.
Reply: thank you – we found similar results for L3 and T12 in this analysis. Thus we now deleted any statements regarding the better use of L3.
- Although the participants were divided into 2 groups (high SMI and low SMI) with a cutoff of lowest quartiles, are there linear correlations or non-linear correlations (e.g. cubic spline curve) between thoracic SMI and the risk of outcomes? It is interesting and meaningful to perform the analyses with SMI as a continuous variable, not only as a binary variable.
Reply: thank you – we agree and now show also data using SMI as a continuous predictor. For the subgroup analysis (treatment response), we still need to focus on 2 groups (i.e., high vs low). Results remain very similar as shown in Table 3 (right column).
- Regarding the analyses in Table 4, the background characteristics of control and intervention groups in high or low SMI groups are not considered randomized because this is the post-hoc analysis of RCT. Therefore, “adjusted” values are necessary, as same as those in Table 3.
Reply: thank you –as suggested by the reviewer, we now also adjusted the analysis for the same confounders as used in the prognostic model and confirm the results. This has been changed in the text.
- In Table 4, what the “interaction” indicates? If it means interaction between intervention and SMI groups, it is surprising that there is a significant interaction despite of the similar OR in high and low SMI groups (0.55 vs. 0.40).
Reply: thank you – we checked again the analysis and found this to be a mistake. Indeed the p value of the interaction analysis is only 0.55. The reported p value was for the OR within the subgroup. Thus, this has now been changed including all statements in this regard. OF note – we now adjusted the analysis based on your comment below and thus all ORs and p values have now been changed.
- Regarding Supplementary Table 1, analyses should also be performed in the patients with NRS < 4 and age < 80 years. Additionally, interaction between SMI and subgroups should be calculated.
Reply: thank you – we agree and have now added the data to the manuscript.
Minor concerns and questions for authors:
- In the Abstract – please insert “(SMI)” after “skeletal mass index” in the first line.
Reply: thank you – we agree and added this as suggested.
- In the Discussion section, the authors described that “in the EFFORT study, there were more routine CT scans with T12 done compared to L3 CT scan.” This statement is closely related to the aim of this study. Please clarify the number of patients performed T12 and L3 scans, respectively.
Reply: thank you – we agree and added these numbers as suggested in abstract ans main text (“Results: 663 of 2028 patients from the EFFORT trial had available CT scans for T12 and 519 among them also had available L3 scans.”).
- Additionally, the authors should describe whether the issue above can be generalized to other populations with references.
Reply: thank you – there are only few studies comparing L3 and T12, and thus generalization to other populations may not be possible at this point. We now added a statement in this regard to the conclusions “There is need for similar research and other patient populations to understand the added value of skeletal muscle index measured at the thoracic vertebra for patient assessment.”.